# RA-XII Suppresses the Development and Growth of Liver Cancer by Inhibition of Lipogenesis via SCAP-dependent SREBP Supression

**DOI:** 10.3390/molecules24091829

**Published:** 2019-05-12

**Authors:** Di Guo, Yurong Wang, Jing Wang, Lihua Song, Zhe Wang, Bingyu Mao, Ninghua Tan

**Affiliations:** 1State Key Laboratory of Natural Medicines, Department of TCMs Pharmaceuticals, School of Traditional Chinese Pharmacy, China Pharmaceutical University, 211198 Nanjing, China; guodi33@163.com (D.G.); yurong1987213@163.com (Y.W.); 18851107621@163.com (J.W.); songlihua4835@163.com (L.S.); wangzhe153807105@163.com (Z.W.); 2State Key Laboratory of Phytochemistry and Plant Resources in West China, Kunming Institute of Botany, Chinese Academy of Sciences, 650201 Kunming, China; 3State Key Laboratory of Genetic Resources and Evolution, Kunming Institute of Zoology, Chinese Academy of Sciences, 650223 Kunming, China; mao@mail.kiz.ac.cn

**Keywords:** RA-XII, lipogenesis, SCAP, SREBP, liver cancer

## Abstract

Lipogenesis plays a critical role in the growth and metastasis of tumors, which is becoming an attractive target for anti-tumor drugs. RA-XII, one of the cyclopeptide glycosides isolated from *Rubia yunnanensis*, exerts anti-tumor effects on liver cancer. However, the underlying mechanisms are not clear. In the present study, the effects of RA-XII on lipogenesis were evaluated and the underlying mechanisms were investigated. The results indicated that RA-XII strongly inhibited tumor growth and lipogenesis (triglycerides and lipid droplets) in HepG2 cells, and the expression of key factors involved in lipogenesis (SREBP, SCD, FASN) was also obviously downregulated. Further investigation showed that the anti-tumor effects of RA-XII were attenuated by SREBP knockdown. Moreover, RA-XII downregulated the expression of SREBP cleavage-activating protein (SCAP), an upstream regulator of SREBP, and siRNA of SCAP prevented its restrained effects on tumor growth and lipogenesis. In addition, the in vivo experiment showed that RA-XII strongly restrained the lipogenesis and growth of liver tumor in nude mice xenograft model. Taken together, these results indicate that RA-XII suppresses the liver cancer growth by inhibition of lipogenesis via SCAP-dependent SREBP suppression. The findings reveal the potentials of RA-XII to be used in a novel therapeutic approach for treating liver cancer.

## 1. Introduction

Liver cancer is one of the most malignant cancers, which affects more than half a million individuals worldwide per year [1], although 5-fluorouracil has been used as the first line treatment in combination with cis-platinum. The continuous decline in cancer death rates over two decades has resulted in an overall drop of 25%. In contrast to the declining trends for other cancers, the death rate is still increasing for liver cancer by almost 3% per year, while the death rate of pancreatic cancer is slightly increasing by 0.3% per year in men but has leveled off in women [2]. Due to the difficulties in early diagnosis, rapid progression, and metastases, the survival rate of liver cancer is extremely low [3]. Hence, there is an urgent need for a novel therapy in liver cancer. In the era of chemotherapeutic treatments, there is a growing interest in targeting metabolic pathways [4]. The tumor growth phase is characterized by a high metabolic demand. An increase in lipogenesis was observed in various cancers, such as breast cancer, colorectal cancer, prostatic cancer, and liver cancer, which is essential to maintain the growth rate of tumor cells, and inhibitors of lipid synthesis have been shown to inhibit cancer cell proliferation [5,6,7,8]. Therefore, inhibiting lipogenesis in tumor cells emerges as a potential therapeutic approach for effective treatment of cancer. However, only a few compounds have been reported to show anti-tumor activity by inhibiting lipogenesis [9,10].

Inhibiting lipogenesis is recognized to treat cancer. Moreover, the sterol-regulatory element binding proteins (SREBP) pathway has been implicated to play an important role in tumor cells growth and lipogenesis [11]. SREBPs are first synthesized as inactive precursors and form a complex with SREBP cleavage-activating protein (SCAP) in endoplasmic reticulum (ER). When cellular sterol level declines, SCAP dissociates from ER and escorts SREBPs to the Golgi apparatus. SREBPs then undergo protease cleavages to liberate the active N-terminal transcriptional domain, which enters the nucleus and activates genes for lipid biosynthesis and uptake [12,13,14]. SCAP has been increasingly recognized as a central regulator in the tumor cells lipogenesis. It has been shown that inhibition of SCAP significantly reduces the lipogenesis and tumor cells growth [15,16]. To the best of our knowledge, few compounds have been shown to demonstrate anti-tumor activity by inhibiting SCAP.

Plant-derived compounds, such as vinblastine and vincristine isolated from *Catharanthus roseus*, irinotecan isolated from *Camptotheca acuminata* and paclitaxel isolated from *Taxus brevifolia*, have been widely used as anti-cancer drugs in the clinic [17,18]. RA-XII (Cyclo(d-alanyl-l-alanyl-*N*,*O*-dimethyl-l-tyrosyl-l-alanyl-*N*-methyl-l-tyrosyl-*O*-β-d-glucopyranosyl-3-hydroxy-*N*-methyl-l-tyrosyl), cyclic (5→6^3^)-ether), bicyclic hexapeptidic glucoside (chemical structure is shown in Figure 1A) isolated from the *Rubia* plants [19,20], has been shown to induce anti-proliferation activity against 11 cancer cell lines [21]. Our previous studies also indicated that RA-XII exerts anti-migration effects on breast cancer cells and inhibits the protective autophagy of liver cancer cells [22,23]. However, the underlying mechanisms are not yet fully understood. In the present study, the effects of RA-XII on lipogenesis in liver cancer cells were evaluated, and the underlying mechanisms were investigated. The results showed that RA-XII strongly inhibited liver tumor growth and lipogenesis, in which SCAP-dependent SREBP suppression was involved. Furthermore, the results demonstrated that RA-XII exerted promising anti-tumor activity and anti-lipogenesis effects in the nude mice xenograft model.

## 2. Results

### 2.1. RA-XII Induces Cell Death and Cell Cycle Arrest in Liver Cancer Cells

Firstly, the effect of RA-XII on the viability of liver cancer cells was examined. RA-XII inhibited the growth of HepG2 cells in a dose- and time-dependent manner (Figure 1B), and showed the same effects on other two liver cancer cells (SMMC7721 and BEL7402 cells, Appendix A). The results indicate that RA-XII exerts a much smaller anti-proliferation effect on HUVEC and HELF cells compared to HepG2, SMMC7721, and BEL7402 cells (Figure 1B, Appendix A). Moreover, RA-XII (0.5–2 μM) dose-dependently induced G2/M cell cycle arrest (Figure 1C,D), and inhibited the expression of cell cycle regulatory proteins Cyclin-dependent kinase 1 (Cdc2), M-phase inducer phosphatase 3 (cdc25C), and G2/mitotic-specific cyclin-B1 (cyclin B1) in HepG2 cells (Figure 1E,F). 

### 2.2. RA-XII Inhibits the Lipid Synthesis in HepG2 Cells 

In normal tissues lipids come from circulating lipids, while cancer cells mainly use *de novo* synthesized lipids. Increased lipogenesis has been proposed to play a key role in cancer cell survival and progression. Pyruvate formed through glycolysis is transformed to citrate in the mitochondria. The ATP citrate lyase (ACL) forms acetyl CoA from citrate. The acetyl CoA carboxylase (ACC) transforms the acetyl CoA into malonyl CoA. Then, the palmitate was synthesized by the fatty acid synthase (FASN) from acetyl CoA and malonyl CoA, and then elongated. Palmitate and stearate are subsequently desaturated by the stearoyl CoA desaturase (SCD) to form palmitoleate and oleate, respectively. These mono-unsaturated fatty acids are preferentially integrated into phospholipids (PL) for membrane synthesis or triglycerides (TG) for exportation in low-density lipoprotein (LDL). 

To determine whether RA-XII has an influence on lipid metabolism of cancer cells, HepG2 cells were treated with RA-XII for 24 h. As shown in the results, RA-XII (2 μM) obviously prevented the lipid synthesis, with decreased amount of total cholesterol (TC), TG, LDL, and lipid droplets in HepG2 cells (Figure 2A,B). Because of the great changes in cell morphology and density, RA-XII treatment seemed to exert better effects in the Oil Red O assay. The regulation of lipid synthesis involves modulation of multiple lipogenic genes at both transcriptional and posttranscriptional levels. Therefore, the expression levels of key lipogenic genes and proteins (FASN and SCD) were examined. RA-XII (0.5–2 μM) dose-dependently down-regulated the gene expression of FASN and SCD. And these observations were confirmed at the protein levels by Western Blot assay in HepG2 cells (Figure 2C–E). Taken together, these data indicate that RA-XII prevents lipogenesis of cancer cells via inhibiting lipogenic genes and proteins.

### 2.3. SREBP-1 Suppression Is Involved in RA-XII-Induced Cell Death and Cell Cycle Arrest in HepG2 Cells

SREBP-1 is a known transcription factor of lipogenic genes, which plays important roles in regulating *de novo* lipogenesis. Moreover, accumulating evidences indicate that SREBP-1 is involved in tumorigenesis. As indicated in Figure 3A, RA-XII (0.5–5 μM) dose-dependently inhibited the expression of SREBP-1 protein in HepG2 cells. The expression of SREBP-1 protein was significantly decreased after HepG2 cells were transfected with SREBP-1 siRNA for 24 h or 48 h (Figure 3B). In this case, SREBP-1 knockdown by siRNA blocked RA-XII-mediated cell death (Figure 3C), G2/M cell cycle arrest (Figure 3D,E), and depression of cell cycle regulatory proteins (Cdc2, cdc25C, cyclin B1) (Figure 3F,G) in HepG2 cells, indicating that SREBP-1 is involved in the cell cycle arrest and cell death induced by RA-XII. 

### 2.4. SCAP-dependent SREBP-1 Suppression Is Involved in RA-XII-induced Cell Death in HepG2 Cells

Previous studies have established the protein SCAP as an upstream regulator of SREBP-1 involved in tumor lipid metabolism. The increased expression of SCAP leads to SREBP-1 activation. RA-XII (0.5–5 μM) dose-dependently inhibited the expression of SCAP protein in HepG2 cells (Figure 4A). The expression of SCAP protein was significantly decreased after HepG2 cells were transfected with SCAP siRNA for 48 h (Figure 4B). However, pretreatment with RA-XII cannot inhibit lipid synthesis (TG, lipid droplets) (Figure 4F,G) and lipid synthesis-related proteins (SREBP-1, FASN, SCD) (Figure 4C–E) in HepG2 cells after SCAP siRNA transfection. SCAP knockdown by siRNA also partially blocked RA-XII-induced cell death (Figure 4H) in HepG2 cells, demonstrating that RA-XII-mediated SREBP suppression and anti-tumor effects are SCAP-dependent.

### 2.5. RA-XII Exerts Anti-tumor and Lipogenesis Inhibition Effects on Xenograft Mouse Model 

To confirm the effect of RA-XII on tumor growth and lipogenesis, the in vivo efficacy was examined with HepG2 xenograft model. RA-XII (20–40 mg/kg) treatment suppressed liver tumor growth (Figure 5A–C). The control group took 15 days to reach the average tumor volume of 1253.19 mm^3^ and nine times in the initial tumor volume (RTV = 9.02), whereas the RTV of the RA-XII treatment at dosage of 40 mg/kg was only 3.36. When determining the inhibitory effect of RA-XII, it was found that RA-XII significantly suppressed tumor growth with the maximal T/C value of 0.37 (mean RTV: 40 mg/kg RA-XII vs. control group) and did not cause significant body weight loss (Figure 5D) of the host mice or other side effects such as mortality, lethargy, and pathologic change of organs (Appendix A). To further explore the in vivo molecular mechanism of RA-XII, Oil Red O staining and TC, TG, LDL, high-density lipoprotein (HDL) levels were evaluated in solid tumors. Consistent with the results of the in vitro study, RA-XII dramatically suppressed the synthesis of TC, TG, LDL, lipid droplets in liver tumors (Figure 6A,B), while it slightly promoted the synthesis of HDL (Figure 6C). Moreover, RA-XII (20–40 mg/kg) dose-dependently reduced the expression of lipid synthesis related genes (FASN, SCD) and proteins (Figure 6D–F). FASN synthesizes palmitate from acetyl CoA and malonyl CoA. Palmitate and stearate are subsequently desaturated by the SCD. These data show that RA-XII can suppress the in vivo tumor growth through inhibiting the SCAP–SREBP pathway and inhibiting critical members involved in lipogenesis.

## 3. Discussion

This study aimed to reveal whether RA-XII inhibits liver tumor growth by suppressing lipogenesis, and to clarify the underlying molecular mechanisms. The present results demonstrated the role of SCAP-dependent SREBP suppression in RA-XII-induced anti-tumor and anti-lipogenesis effects. In addition, the anti-tumor and anti-lipogenesis effects of RA-XII in vivo were confirmed. 

In the present study, the findings first show that RA-XII inhibited the cell viability and induced G2/M cell cycle arrest in HepG2 cells in a lipid-deficient environment (Figure 1C,D). The cell cycle is an intricate process implicated in cell growth and division, DNA-damage response, and diseases including cancer [24,25]. Cyclins and cyclin-dependent protein kinases (Cdks) are key regulators of the cell cycle progression. Herein, an obvious decrease in these proteins was observed (Figure 1E,F). However, the block effect of RA-XII on cell cycle progression is maybe reversible; cells would either undergo repair mechanisms or follow the death pathway when the arrest of cell cycle progression at G2/M phase occurs.

Abnormal lipid metabolism plays important roles in cancer development and progression. It has been reported that some small molecules and genes exert anti-tumor effects due to their effects on lipid metabolism and G2/M cell cycle arrest [26,27]. Elevated fatty acid synthesis is one of the most important alterations of cancer cell metabolism [28,29,30]. Previous studies have found that many cancer cells show high rates of *de novo* lipid synthesis, especially in liver cancer. It is noteworthy that the liver is the epicenter of lipid metabolism where most cholesterol and fatty acids (FAs) are synthesized [8]. Accumulation of lipid droplets and triglycerides are two frequently observed phenotypes in lipogenesis and manifestations of tumor abnormal lipid metabolism [9,10]. Moreover, overexpression of genes encoding lipogenic enzymes was responsible for the elevated lipid biosynthesis in cancer. In fact, overexpression of lipogenic enzymes is reported as a common characteristic of many cancers, and inhibition of different enzymes within the fatty acid biosynthetic pathway can block cancer cell growth [31,32,33]. Among them, SCD and FASN are two important enzymes during cell *de novo* lipogenesis. The significant roles of SCD and FASN in cancer development have been well established in the past [34,35]. Elevated expression of SCD and FASN in cancer cells is related to markedly worse prognosis in many human cancers, including liver cancer. The SCD inhibitor BZ36 and FASN inhibitor C75 have shown anti-tumor effects in a pre-clinical xenograft model [36,37]. In this study, the results show that RA-XII significantly suppressed accumulation of lipid droplets and triglycerides in HepG2 cells (Figure 2A,B). The expression of SCD and FASN were also down-regulated after RA-XII treatment (Figure 2C–E). Although expression levels of FASN and SCD reach a minimum at 1–2 μM of RA-XII treatment, sulforhodamine B (SRB) assay shown in Figure 1B indicates that the peak inhibitory effect of RA-XII was observed at a conc of 5 μM or slightly higher. Taken together, the results demonstrate that the anti-tumor effects of RA-XII in liver cancer cells are, at least partly, the results of suppressed lipogenesis. 

Then the mechanisms underlying RA-XII suppresses tumor lipogenesis were investigated. SREBPs are a family of transcription factors that regulate lipid homeostasis by regulating the expression of the core and rate-limiting enzymes involved in lipid synthesis, including SCD and FASN [38]. Moreover, the activities of SREBP are regulated by SREBP cleavage-activating protein (SCAP). It has been reported that elevated expression of SCAP–SREBP pathway has been detected in several cancer types and was closely correlated with malignant transformation, cancer progression, and metastasis [14,39]. Inhibition of the SCAP–SREBP pathway leads to impaired tumor growth [13,40,41]. In this study, the results indicate that RA-XII robustly decreased SCAP and SREBP activation. Moreover, the exposure to SCAP and SREBP siRNAs could block the anti-tumor and anti-lipogenesis effects of RA-XII in HepG2 cells (Figure 3 and Figure 4), suggesting that anti-tumor and anti-lipogenesis effects of RA-XII may be mediated through down-regulation of the SCAP–SREBP pathway. 

We previouly reported that RA-XII suppressed protective autophagy and promoted apoptosis in HepG2 cells. RA-XII treatment substantially reduced both autophagosomes and autophagic flux through activation of mTOR signaling pathway [23]. Herein, the new results indicated that RA-XII’s anticancer properties can also be attributed to its inhibition effects on cancer lipid metabolism. The other role of RA-XII in cancer remains to be investigated in our further study.

In order to assess the bioactivity of RA-XII in vivo, the nude mice xenograft model was established. The results showed that RA-XII significantly inhibited the tumor growth (Figure 5) and decreased the accumulation of lipid droplets and triglycerides in solid tumors (Figure 6A–C). The expression of proteins involved in the SCAP–SREBP pathway were also obviously downregulated. These effects of RA-XII on the xenograft model were in accordance with its lipogenesis inhibition effects in vitro. 

## 4. Materials and Methods 

### 4.1. Reagents 

RA-XII was isolated from the roots of *Rubia yunnanensis*. The details of extraction and isolation procedures for RA-XII have been reported previously [19,20], and its purity has been shown in Appendix A. RA-XII powder was dissolved in dimethyl sulphoxide (DMSO) (Sigma, St. Louis, MO, USA) at 20 mM as a stock solution stored at −20 °C. The stock was diluted with DMSO to a thousand times the final concentration and reconstituted in MEM medium prior to use. The control HepG2 cells were treated with the same concentration of DMSO. Sulforhodamine B was purchased from Sigma (St. Louis, MO, USA). Anti-SCAP and cyclin B1 antibodies were purchased from Cell Signaling Technology (Beverly, MA, USA). Anti-Cdc2 and anti-cdc25C antibodies were purchased from Proteintech Group (Rosemont, PA, USA). Anti-β-actin and SREBP-1 antibodies were purchased from Santa Cruz Biotechnology (Dallas, TX, USA). PI cell cycle analysis kit was purchased from KeyGen Biotech. Co. Ltd. (Nanjing, Jiangsu, China). 

### 4.2. Cell Culture 

Human liver cancer HepG2 cells were purchased from Type Culture Collection of Chinese Academy of Sciences, Shanghai, China in 2017 and cultured in MEM complete medium (supplemented with 10% FBS, 100 U/mL penicillin and 100 μg/mL streptomycin). HepG2 cells at passages 3–15 and approximately 70% confluent were seeded onto 6-well or 24-well plate for further experiments. Before RA-XII treatment, cells were grown in MEM medium with 1% lipoprotein deficient serum for 6 h.

### 4.3. Cell Viability Assay

Cell viability was determined via sulforhodamine B (SRB) assay. HepG2 cells were seeded in 96-well plates at a density of 1 × 10^4^ cells and incubated for 24 or 48 h. After treatment with RA-XII, 100 µL of 10% TCA was added to each well and fixed in for 1 h. After washing three times with water, 100 µL of 4% SRB was added to each well and stained 15 min. Then, washed three times with 1% TCA, stained cells were dissolved with 10 mM unbuffered Trisbase (pH = 10.5) and were measured the absorbance at 540 nm using a microtitre plate reader (Bio-Rad, Hercules, CA, USA).

### 4.4. Cell Cycle Analysis 

Cell cycle distribution in HepG2 cells was assessed by propidium iodide (PI) staining. PI binds to DNA by intercalating between the bases with the little or no sequence preference. Consequently, the degree of fluorescence is proportional to the amount of cellular DNA, which itself is indicative of cell cycle phase (as cells progress through the cell cycle, the amount of DNA ultimately doubles). PI will also label RNA, so the addition of RNase A is necessary for accurate determination of the percentage of cells in each phase of the cell cycle. 

After treatment with RA-XII, the attached cells were harvested, washed with PBS and fixed in 70% (*v*/*v*) ethanol at 4 °C overnight. Then the cells were resuspended in 500 μL of staining solution containing PI and RNase A at 37 °C and analyzed using flow cytometer (Attune NxT, Thermo, Rockford, IL, USA). The PI fluorescence signal peak versus the integral was used to discriminate among the S, G0/G1 and G2/M phases of the cell cycle using flowjo software (Tree Star, Ashland, OR, USA).

### 4.5. Western Blot Analysis

Cells and tumor tissues were lysed in RIPA lysis buffer (Beyotime, Nanjing, Jiangsu, China). The homogenates were centrifuged at 11600 ×g for 20 min. Total protein concentration of the supernatants was assessed by BCA kit (Thermo, Rockford, IL, USA). Equal amounts of lysates were separated on 8–15% SDS-polyacrylamide gel electrophoresis and electrophoretically transferred onto polyvinylidenedifluoride membranes (PVDF) (Millipore, Bedford, MA, USA). Membranes were then blocked with 5% BSA in Tris-buffed-saline with Tween (TBST) for 1 h, followed by incubation with diluted primary antibodies (overnight, 4 °C). Membranes were washed with 0.1% Tween-20 in Tris-buffered saline (TBS) and incubated with horseradish peroxidase-conjugated secondary antibodies for 1 h at room temperature. The immunoreactive proteins were then detected by the ECL-Plus Western Blotting Detection System (5200Multi, Tacon, Shanghai, China).

### 4.6. Preparation of Cell Membrane Fractions and Nuclear Extracts

Cell membrane fractions were isolated using ProteoExtract Transmembrane Protein Extraction Kit (Novagen, Darmstadt, Germany) and nuclear fractions were isolated with NE-PER Nuclear and Cytoplasmic Extraction Reagents Commercial Kits (Thermo, Rockford, IL, USA) according to manufacturer’s specifications.

### 4.7. Quantitative PCR

Total RNA was extracted from cells or tumor samples using Trizol reagent (Invitrogen, Camarillo, CA, USA). Isolated RNA was reverse-transcribed into cDNA using a cDNA synthesis kit (Vazyme, Nanjing, Jiangsu, China) following standard protocols. Quantitative PCR (qPCR) was performed using synthetic primers and SYBR Green (Thermo, Rockford, IL, USA) with a IQ5 Detection System. After incubation at 50 °C for 2 min and 95 °C for 10 min, samples were subjected to 40 cycles of 95 °C for 15 s and 60 °C for 1 min. qPCR primers were listed in Table 1. The final results were all normalized as fold change of the target gene/GAPDH.

### 4.8. Oil Red O Staining

Before staining, cells were washed with PBS and fixed in 4% paraformaldehyde for 1 h. Tumor tissues were imbedded with OCT Tissue-Tek compound (Sakura Finetek, GA, USA) onto a round cork plate and rapidly frozen to −30 °C. By using a cryotome (Microm HM 550, Walldorf, Germany), the frozen block was serially sectioned into 20 μm slices (at least 10 slices per block), which were thaw-mounted on glass slides. Then the fixed cells or frozen sections were stained with Oil Red O solution as previously described [10,41] and photographed under the light microscope (DMi8, Leica, Solms, Germany) at magnifcation of 200×.

### 4.9. Measurement of TC, TG, LDL and HDL

After treatment with RA-XII, HepG2 cells were harvested, washed with PBS, and lysed in 1% Triton X-100 lysis buffer (Beyotime, Nanjing, Jiangsu, China) for 30 minutes. Tumor tissues were weighted and homogenized at 2500 rpm/min for 10 min, then centrifuged, and the supernatants were harvested. Protein concentration was assessed by BCA kit (Thermo, Rockford, IL, USA). TC, TG, LDL and HDL in the HepG2 cell lysates or solid tumors were assayed using commercial kits (A111-1, A110-1, A113-1, A112-1, Jiancheng, Nanjing, China) according to manufacturer’s specifications. This study set blank wells, standard wells, and detected sample wells. The blank wells were added with sample diluent and other wells were added with standard preparation or detected samples, respectively. For TC and TG detection, all wells were then filled with working solution at 37 °C for 10 min and Optical density (OD) values were measured at 510 nm. For LDL and HDL detection, all wells were then filled with working solution A and working solution B at 37 °C for 5 min successively and OD values were measured at 546 nm two times. 

### 4.10. SCAP and SREBP Knockdown in HepG2 Cells by siRNA

HepG2 cells cultured in 24-well plates were transfected with NC siRNA (native control siRNA with scrambled sense), SREBP-1siRNA, or SCAP siRNA (Genepharma, Shanghai, China) when cells reached 50% confluence. siRNA and Lipofectamine 3000 (Invitrogen, Camarillo, CA, USA) were premixed in OPTI-medium (Invitrogen, Camarillo, CA, USA) according to the manufacturer’s instructions and then applied to the cells. After 24 h transfection, OPTI-medium was replaced by complete MEM medium. Then 6 h later, HepG2 cells were treated with RA-XII. All siRNA sense strands were listed in Table 2.

### 4.11. Xenograft Mouse Model

Female Balb/c nude mice, 6–8 weeks old, weighing 18–22 g, were purchased from Shanghai Laboratory Animal Center, Chinese Academy of Sciences. The mice were raised in air-conditioned rooms under controlled lighting (12 h light per day) and provided with food and water at discretion. All the animal experimental procedures were performed strictly according to protocols approved by the Animal Care Committee of China Pharmaceutical University (permission number SYXKSU 2016-0011). All animals were treated and used in a scientifically valid and ethical manner. 

Viable HepG2 cells (3 × 10^6^/100 μL PBS per mouse) were subcutaneously (s.c.) injected into the right flank of the nude mice. When the average s.c. tumor volume reached 100–160 mm^3^, animals were randomly divided into 4 groups (*n* = 6) and intravenously administered the following regimens thrice weekly: (a) vehicle; (b) RA-XII (10 mg/kg); (c) RA-XII (20 mg/kg); (d) RA-XII (40 mg/kg). Tumor size and body weight were recorded three times a week with a caliper (calculated volume = shortest diameter^2^ × longest diameter/2). After 15 days, mice were sacrificed and solid tumors were isolated for further analysis.

### 4.12. Statistical Analysis 

Statistical significance was determined using the GraphPad Prism 5 software (San Diego, CA, USA). Data are presented as the mean ± SEM. Statistical analyses among multiple groups were performed using one-way ANOVA followed by Bonferroni’s post hoc test. A value of *p* < 0.05 was considered to be statistically significant. 

## 5. Conclusions

These findings demonstrate that RA-XII significantly suppresses the cell growth and lipogenesis in HepG2 liver cancer cell line. The main mechanism may be inhibition of SCAP dependent SREBP signaling pathway. Moreover, RA-XII exerted anti-tumor and anti-lipogenesis effects in the nude mice liver xenograft model. These results indicate that RA-XII might be considered as a novel therapeutic agent for treating liver cancer in the future.

## Figures and Tables

**Figure 1 molecules-24-01829-f001:**
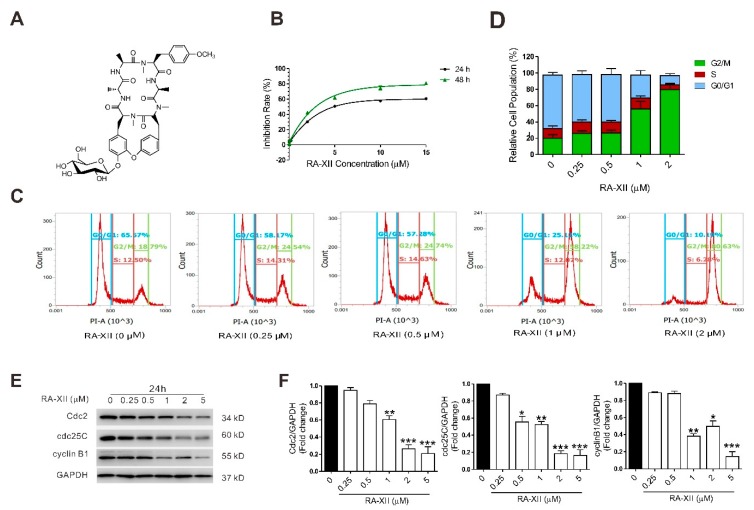
RA-XII induces cell death and G2/M cell cycle arrest in HepG2 cells. (**A**) Chemical structure of RA-XII. (**B**) Sulforhodamine B (SRB) measurement of cell vitality. (**C**) Flow cytometry of cell cycle distributions. (**D**) Statistical graph of cell cycle distributions. (**E**) Protein expression of Cdc2, cdc25C, and cyclin B1. (**F**) Statistical graphs of protein expression. Cells were treated with different concentrations of RA-XII for 24 h. Results are means ± SEM of three independent experiments. * *p* < 0.05, ** *p* < 0.01, *** *p* < 0.001, compared with 0 μM.

**Figure 2 molecules-24-01829-f002:**
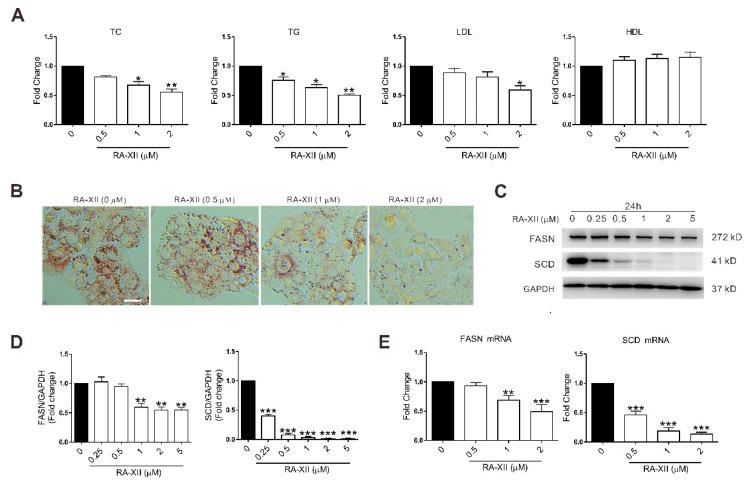
RA-XII inhibits the lipid synthesis in HepG2 cells. (**A**) Concentrations of total cholesterol (TC), triglycerides (TG), low-density lipoprotein (LDL), and high-density lipoprotein (HDL). (**B**) Lipid droplet test by Oil Red O staining. Scale bar: 50 μm. (**C**) Protein expression of the (fatty acid synthase) FASN and stearoyl CoA desaturase (SCD). (**D**) Statistical graphs of protein expression. (**E**) Gene expression of FASN and SCD. Cells were treated with different concentrations of RA-XII for 24 h. Results are means ± SEM of three independent experiments. * *p* < 0.05, ** *p* < 0.01, *** *p* < 0.001, compared with 0 μM.

**Figure 3 molecules-24-01829-f003:**
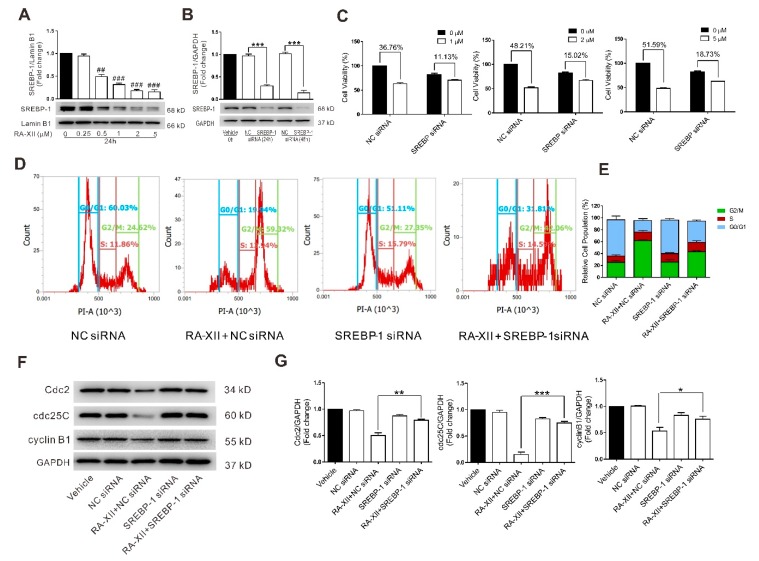
SREBP-1 supression is involved in RA-XII-induced cell death and cell cycle arrest in HepG2 cells. (**A**) Protein expression of SREBP-1 and the statistical graph. Cells were treated with different concentrations of RA-XII for 24 h. (**B**) Protein expression of SREBP-1 after HepG2 cells were transfected with 40 nM SREBP-1 siRNA for 24 or 48 h. (**C**) SRB measurements of cell vitality. Cells were transfected with 40 nM SREBP-1 siRNA or NC siRNA for 24 h, followed by treatment with the indicated concentrations of RA-XII for 24 h. (**D**) Flow cytometry of cell cycle distributions. (**E**) Statistical graph of cell cycle distributions. (**F**) Protein expression of Cdc2, cdc25C and cyclin B1. (**G**) Statistical graphs of protein expression. Cells were transfected with 40 nM SREBP-1 siRNA or NC siRNA for 24 h, followed by treatment with 1 μM RA-XII for 24 h (**D** and **F**). Results are means ± SEM of three independent experiments. ^##^
*p* < 0.01, ^###^
*p* < 0.001, compared with 0 μM. * *p* < 0.05, ** *p* < 0.01, *** *p* < 0.001.

**Figure 4 molecules-24-01829-f004:**
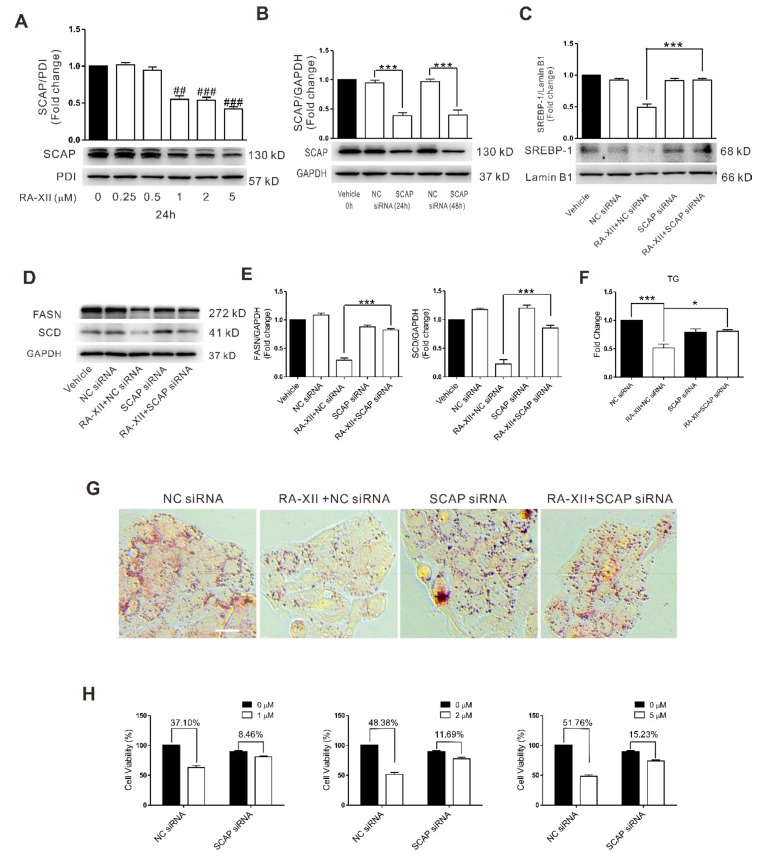
SREBP cleavage-activating protein (SCAP) dependent SREBP-1 supression is involved in RA-XII-induced cell death in HepG2 cells. (**A**) Protein expression of SCAP and the statistical graph. Cells were treated with different concentrations of RA-XII for 24 h. (**B**) Protein expression of SCAP after HepG2 cells were transfected with 40 nM SCAP siRNA for 24 or 48 h. (**C**) Protein expression of SREBP-1. (**D**) Protein expression of FASN and SCD. (**E**) Statistical graphs of protein expression. (**F**) Concentrations of TG. (**G**) Lipid droplet test by Oil Red O staining. Scale bar: 50 μm. (**H**) SRB measurements of cell vitality. Cells were transfected with 40 nM SCAP siRNA or NC siRNA for 48 h, followed by treatment with 1 μM RA-XII (**C**–**G**) or the indicated concentrations of RA-XII (**H**) for 24 h. Results are means ± SEM of three independent experiments. ^##^
*p* < 0.01, ^###^
*p* < 0.001, compared with 0 μM. * *p* < 0.05, *** *p* < 0.001.

**Figure 5 molecules-24-01829-f005:**
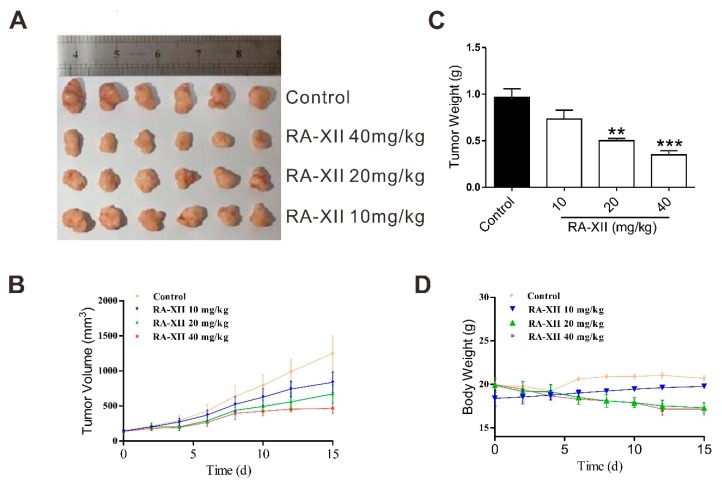
RA-XII exerts anti-tumor effects on the xenograft mouse model. Mice were treated with RA-XII at 10, 20, or 40 mg/kg intravenously thrice a week. After 15 days, mice were sacrificed and solid tumors were isolated. (**A**) Image of isolated tumors at the end of experiment. (**B**) Volume change of tumors. (**C**) Weight of tumors at the end of experiment. (**D**) Body weight change of the mice. Tumor size and body weight were recorded three times a week. *n* = 6. ** *p* < 0.01, *** *p* < 0.001, compared with vehicle control.

**Figure 6 molecules-24-01829-f006:**
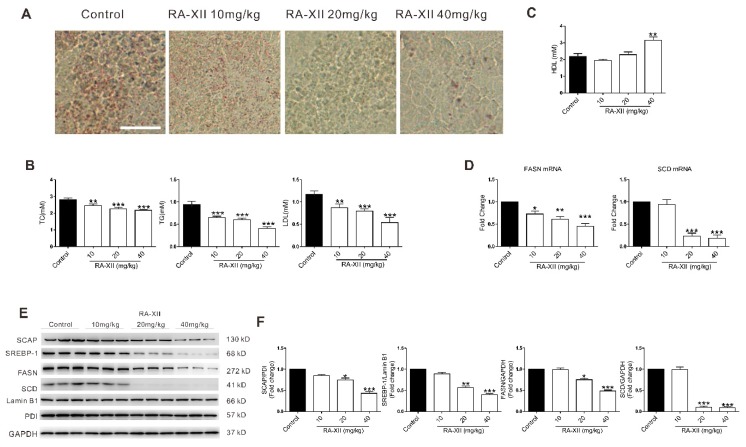
RA-XII exerts lipogenesis inhibition effects on the xenograft mouse model. After treatment with RA-XII for 15 days, mice were sacrificed and solid tumors were isolated. (**A**) Lipid droplet test by Oil Red O staining. Scale bar: 50 μm. (**B**) Concentrations of TC, TG, and LDL. (**C**) Concentrations of HDL. (**D**) Gene expression of FASN and SCD. (**E**) Protein expression of SCAP, SREBP-1, FASN, and SCD. (**F**) Statistical graphs of protein expression. *n* = 6. * *p* < 0.05, ** *p* < 0.01, *** *p* < 0.001, compared with vehicle control.

**Table 1 molecules-24-01829-t001:** Primers for RT-PCR.

Gene	Sense (5′-3′)	Anti-sense (5′-3′)
FASN	CGCCGAGTACAATGTCAACAA	AGTGGGGAGATGAGGGGAGT
SCD	CGACGTGGCTTTTTCTTCTC	GGGGGCTAATGTTCTTGTCA
GAPDH	AATCCCATCACCATCTTCCAG	ATGAGTCCTTCCACGATACCAA

**Table 2 molecules-24-01829-t002:** Sequence of target gene siRNA.

Gene	Sense Strand (5′-3′)
SREBP-1 siRNA	GCAACACAGCAACCAGAAATT
NC siRNA	UUCUCCGAACGUGUCACGUTT
SCAP siRNA	CCUACCUUGUGGUGGUUAUTT

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
