# Peer review of "RA-XII Suppresses the Development and Growth of Liver Cancer by Inhibition of Lipogenesis via SCAP-dependent SREBP Supression"

_molecules, 2019, doi:10.3390/molecules24091829_

Round 1

Reviewer 1 Report

This paper is a valuable contribution that explores the role of lipogenesis in the growth of liver cancer cells studied in vitro and in vivo (xenografts); several techniques were explored and contains a great deal of information. However, I cannot recommend this manuscript to be published in this version in molecules due to the issues as follows.

Please show evidence that the RA-XII compound extracted was clean and free of impurities: show characterization spectra.

Western Blots are nitid.

Please add the full IUPAC name of the RA-XII compound. 

Please define NC in NC siRNA.

Please define all abbreviations used within the text.

Please explain thoroughly how the fluorescence signal was used to discern among the cell cycle phase of the cells in the "cell cycle analysis" section of M&Ms.

Please explain thoroughly how were TC, TG, LDL and HDL measured?

English corrections are suggested:

Line 30: "indicated" for "indicate"

Line 31: "revealed" for "reveal"

Line 32: "the" for "a"

Line 46: "have been showed" for "have shown"

Line 60: "for" for "as"

Line 61: "was" for "is"

Line 64: "have" for "are"

Line 72: "were" for "was"

Line 73: check font

Line 75: "indicated" for "indicate"

The phrase that begins in line 74 and ends in line 76 is not clear. Please clarify. What cancer cells are you referring to?

Line 77 and 116, 117, 120, 128 and elsewhere: "expressions" for "expression"

Line 96: "indicated" for "indicate"

Line 108: "were" for "was"

Line 171: "whether"

Line 175, 196: "showed" for "show"

Line 189, 196: "triglycerides"

Line 198:"demonstrate"

Line 207: "indicate"

Line 293: "specifications"

Author Response

To Reviewer 1

Point 1: Please show evidence that the RA-XII compound extracted was clean and free of impurities: show characterization spectra.

RESPONSE: Thanks for the valuable comment. The HPLC chromatogram for the purity of RA-XII has been shown in Fig. S3 and described on page 9 (lines 256-258).

Point 2: Western Blots are nitid.

RESPONSE: The western blots of all figures have been re-edited according to the reviewer’s suggestion.

Point 3: Please add the full IUPAC name of the RA-XII compound.

RESPONSE: The full IUPAC name of RA-XII has been added in the Introduction section (page 2, lines 65-66).

Point 4: Please define NC in NC siRNA.

RESPONSE: NC siRNA is native control siRNA with scrambled sense and this issue has been addressed in the M&Ms section on page 11 (lines 341-342) and Table 2.

Point 5: Please define all abbreviations used within the text.

RESPONSE: All abbreviations have been defined in the manuscript.

Point 6: Please explain thoroughly how the fluorescence signal was used to discern among the cell cycle phase of the cells in the "cell cycle analysis" section of M&Ms.

RESPONSE: Cell cycle distribution in HepG2 cells was assessed by propidium iodide (PI) staining and the method has been described in detail in the "cell cycle analysis" section of M&Ms (page 10, lines 282-292).

Point 7: Please explain thoroughly how were TC, TG, LDL and HDL measured?

RESPONSE: Measurement methods of TC, TG, LDL and HDL have been explained thoroughly in the M&Ms section (page 11, lines 328-339).

Point 8: English corrections are suggested:

Line 30: "indicated" for "indicate"

Line 31: "revealed" for "reveal"

Line 32: "the" for "a"

Line 46: "have been showed" for "have shown"

Line 60: "for" for "as"

Line 61: "was" for "is"

Line 64: "have" for "are"

Line 72: "were" for "was"

Line 73: check font

Line 75: "indicated" for "indicate"

Line 77 and 116, 117, 120, 128 and elsewhere: "expressions" for "expression"

Line 96: "indicated" for "indicate"

Line 108: "were" for "was"

Line 171: "whether"

Line 175, 196: "showed" for "show"

Line 189, 196: "triglycerides"

Line 198:"demonstrate"

Line 207: "indicate"

Line 293: "specifications"

RESPONSE: All English corrections above-mentioned have been revised as suggested and highlighted in the manuscript.

Point 9: The phrase that begins in line 74 and ends in line 76 is not clear. Please clarify. What cancer cells are you referring to?

RESPONSE: Thanks and revised (page 2, lines 80-82).

Reviewer 2 Report

@page { margin: 2cm } p { margin-bottom: 0.25cm; line-height: 120% }

In this manuscript (ID molecules-491842), authors Guo et al describe their study of the cyclopeptide glycoside, RA-XII, and elucidation of its anti-cancer properties towards liver cancer.

The authors have attempted to do this by using the liver cancer cell line HepG2 as a model, as well as an in vivo xenograft model in nude mice. The studies in HepG2 cells involve experiments that estimate inhibitory action of RA-XII for tumor growth, lipogenesis, and the expression levels of proteins involved in lipogenesis (SREBP, SCD). Studies using the xenograft model involved looking at tumor size, volume, and lipogenesis. From the results obtained, the authors conclude that RA-XII manifests its anti-liver cancer properties through inhibition of lipogenesis, which in turn works through suppression of SREBP.

Overall, the work is sound, the experiments are performed rigorously, and the conclusions drawn are reasonably justified.

Some syntax and grammatical errors need to be rectified by proof-reading the manuscript rigorously.

A few examples are given below -

Introduction section:

1) lines 46-47 should be: “However, only a few compounds have been reported to show anti-tumor activity by inhibiting lipogenesis”

2) line 57 should be “few compounds have been shown to demonstrate anti-tumor activity by inhibiting SCAP”

3) line 60: it should be “widely used as anti-cancer drugs in the clinic”. Also, remove “one kind of” before “bicyclic hexapeptidic glucoside”. Also, it shold be “chemical structure is shown in Fig. 1A”.

4) line 62: should be anti-proliferation “activity” and not “activities”

5) line 63 sould be “on breast cancer cells” not “of”, line 64-65 should be “the underlying mechanisms are not yet fully understood”

Results section:

1) lines 75 should be: “RA-XII exerts a much smaller anti-proliferative effect on HUVEC and HELF cells compared to HepG2 cells”

2) line 125 should be “previous studies have established the protein SCAP as an upstream regulator”. Line 126, change “tomor” to “tumor”.

3) in many places, change “expressions” to “expression” when referring to proteins.

Discussion section:

line 171, correct “wether” to “whether”.

In general, the figure legends need to be better worded, and more descriptive about what each figure shows. For example, there is no description for Fig. 3C, or it is very vague, “SRB measurement of cell vitality”. No mention about the three doses (0, 2 and 5 uM) of RA-XII.

Results from the SRB assay shown in Fig 1B indicate that the peak inhibitory effect of RA-XII is observed at a conc of 5 uM or slightly higher, at both 24h and 48h.

These seem to be in contrast to other results, for instance expression levels of FASN and SCD, which reach a minimum at 1-2 uM of RA-XII (Fig2 C and D), or level of SREBP-1 expression which reaches a minimum at 2 uM RA-XII (Fig 3A) . These are hard to reconcile with each other, and an explanation for this discrepancy should be provided.

Fig. 2A and D: fold change in levels of TC, TG and LDL in 2A at 2 uM dose of RA-XII seem to be much less, than what is reflected in the oil Red O staining assay (2D) at the same concentration. Again, some explanation needs to be provided for this discrepancy.

Results shown in Fig. 5: First of all, the legend needs to be much more descriptive. For instance, Fig. 5A is described merely as “image of tumors”. That much is obvious. But what are the 6 images in each row? I am assuming those are the six replicates, but this needs to be clarified. Otherwise they could also be mistakenly assumed to be images at different time points. Also, if those are tumors from six different animals, then on what day (between 1-15) are they from?

Author Response

To Reviewer 2

Point 1: Some syntax and grammatical errors need to be rectified by proof-reading the manuscript rigorously.

Introduction section:

1) lines 46-47 should be: “However, only a few compounds have been reported to show anti-tumor activity by inhibiting lipogenesis”

2) line 57 should be “few compounds have been shown to demonstrate anti-tumor activity by inhibiting SCAP”

3) line 60: it should be “widely used as anti-cancer drugs in the clinic”. Also, remove “one kind of” before “bicyclic hexapeptidic glucoside”. Also, it shold be “chemical structure is shown in Fig. 1A”.

4) line 62: should be anti-proliferation “activity” and not “activities”

5) line 63 sould be “on breast cancer cells” not “of”, line 64-65 should be “the underlying mechanisms are not yet fully understood”

Results section:

1) lines 75 should be: “RA-XII exerts a much smaller anti-proliferative effect on HUVEC and HELF cells compared to HepG2 cells”

2) line 125 should be “previous studies have established the protein SCAP as an upstream regulator”. Line 126, change “tomor” to “tumor”.

3) in many places, change “expressions” to “expression” when referring to proteins.

Discussion section:

line 171, correct “wether” to “whether”.

RESPONSE: Thanks and revised carefully in the whole manuscript.

Point 2: In general, the figure legends need to be better worded, and more descriptive about what each figure shows. For example, there is no description for Fig. 3C, or it is very vague, “SRB measurement of cell vitality”. No mention about the three doses (0, 2 and 5 uM) of RA-XII.

RESPONSE: All figure legends have been carefully revised.

Point 3: Results from the SRB assay shown in Fig 1B indicate that the peak inhibitory effect of RA-XII is observed at a conc of 5 uM or slightly higher, at both 24h and 48h.

These seem to be in contrast to other results, for instance expression levels of FASN and SCD, which reach a minimum at 1-2 uM of RA-XII (Fig2 C and D), or level of SREBP-1 expression which reaches a minimum at 2 uM RA-XII (Fig 3A). These are hard to reconcile with each other, and an explanation for this discrepancy should be provided.

RESPONSE: RA-XII maybe induces cell death not only by the lipogenesis pathway but also by the other pathways such as AMPK/mTOR/P70S6K as reported in our previous results (Song et al., Molecules 2017, 22, 1934). This issue has been addressed in the Discussion section on page 9 (lines 225-229).

Point 4: Fig. 2A and D: fold change in levels of TC, TG and LDL in 2A at 2 uM dose of RA-XII seem to be much less, than what is reflected in the oil Red O staining assay (2D) at the same concentration. Again, some explanation needs to be provided for this discrepancy.

RESPONSE: Because of the great changes in cell morphology and density, RA-XII treatment seemed to exert better effects in the Oil Red O assay and the result has been described on page 3 (lines 106-107).

Point 5: Results shown in Fig. 5: First of all, the legend needs to be much more descriptive. For instance, Fig. 5A is described merely as “image of tumors”. That much is obvious. But what are the 6 images in each row? I am assuming those are the six replicates, but this needs to be clarified. Otherwise they could also be mistakenly assumed to be images at different time points. Also, if those are tumors from six different animals, then on what day (between 1-15) are they from?

RESPONSE: All figure legends have been revised carefully. The 6 images in each row are tumors from six different animals at the end of experiment. This issue has been addressed on page 7 (lines 181-186).

Reviewer 3 Report

The reviewed paper at title:

RA-XII suppresses the development and growth of liver cancer by inhibition of lipogenesis via  SCAP-dependent SREBP supression

This study shows the effect of RA-XII on lipogenesis and examined the basic mechanisms. The results indicate that RA-XII strongly inhibits tumor growth and lipogenesis in HepG2 cells, and the expression of key factors involved in lipogenesis (SREBP, SCD, FASN) was also significantly reduced. Further studies have shown that the anti-cancer effects of RA-XII have been weakened by the SREBP knockdown. In addition, RA-XII reduced the expression of SCAP, the SREBP regulator, and SCAP siRNA prevented its limited impact on tumor growth and lipogenesis. In addition, the in vivo experiment demonstrated that RA-XII strongly inhibits lipogenesis and growth of liver cancer in a model in mice.

The paper seems to be acceptable (especially in this journal section) but, in my opinion, it requires some modifications. Additionally, several questions should be answered by the authors in detail, as many important issues are described too superficially:

Line 38-39: I am asking for a more detailed description      related to the liver cancer, for example a comparison can be made with the      pancreatic cancer and even simple pancreatitis.

Line 59-60: I believe that it is worth better to      describe these compounds of plant origin to improve the readability of the      text.

Line 87-91: I think it is worth to better      describe the aspect related to lipogenesis. The fatty acids are released from the      aforementioned substances found in the capillary adjacent to the      adipocytes due to the enzyme lipoprotein lipase. Then, the proper      synthesis takes place thanks to three other enzymes - acetyl-CoA synthase,      glycerol phosphate acyltransferase and phospholipid phosphohydrolase.

Line 119: I am asking for more precise      characteristics related to the cytometric method.

Line 156-157: I am asking for clarification of      the text regarding the reduction of gene expression.

Line 190-192: I am asking for precise specification      of the topic in the field of gene overexpression and lipid biosynthesis.

In conclusion, the paper seems to be acceptable but requires some revisions. The whole layout and neatness of the paper do not leave too much objections, as it is prepared very carefully, but the quality of the discussion requires several amendments. Please answer all my questions and comments and attach the manuscript with marked changes. The objections presented by me do not undermine the quality of the paper, which will support in the further publishing process, certainly after careful consideration of my comments.

Author Response

To Reviewer 3

Point 1: Line 38-39: I am asking for a more detailed description related to the liver cancer, for example a comparison can be made with the pancreatic cancer and even simple pancreatitis.

RESPONSE: Thanks for the constructive suggestion. Liver cancer is one of the most malignant cancers, which affects more than half million individuals worldwide per year, although 5-fluorouracil has been used as the first line treatment in combination with cis-platinum. The continuous decline in cancer death rates over two decades has resulted in an overall drop of 25%. In contrast to the declining trends for other cancers, the death rate is still increasing for liver cancer by almost 3% per year, while the death rate of pancreatic cancer is slightly increase by 0.3% per year in men but has leveled off in women. This issue about liver cancer has been addressed in the Introduction section (page 1, lines 36-41).

Point 2: Line 59-60: I believe that it is worth better to describe these compounds of plant origin to improve the readability of the text.

RESPONSE: The plant origins of these compounds have been added on page 2, lines 62-64.

Point 3: Line 87-91: I think it is worth to better describe the aspect related to lipogenesis. The fatty acids are released from the aforementioned substances found in the capillary adjacent to the      adipocytes due to the enzyme lipoprotein lipase. Then, the proper synthesis takes place thanks to three other enzymes - acetyl-CoA synthase, glycerol phosphate acyltransferase and phospholipid phosphohydrolase.

RESPONSE: In normal tissues lipids come from circulating lipids, while cancer cells mainly use de novo synthesized lipids. Increased lipogenesis has been proposed to play a key role in cancer cell survival and progression. Pyruvate formed through glycolysis is transformed to citrate in the mitochondria. The ATP citrate lyase (ACL) forms acetyl CoA from citrate. The acetyl CoA carboxylase (ACC) transforms the acetyl CoA into malonyl CoA. Then, the palmitate was synthesized by the fatty acid synthase (FASN) from acetyl CoA and malonyl CoA, and then elongated. Palmitate and stearate are subsequently desaturated by the stearoyl CoA desaturase (SCD) to form palmitoleate and oleate, respectively. These mono-unsaturated fatty acids are preferentially integrated into phospholipids (PL) for membrane synthesis or triglycerides (TG) for exportation in low-density lipoprotein (LDL) (page 3, lines 94-102).

Point 4: Line 119: I am asking for more precise characteristics related to the cytometric method.

RESPONSE: Cell cycle distribution in HepG2 cells was assessed by propidium iodide (PI) staining and the method has been described in detail in the M&Ms section (page 10, lines 282-292).

Point 5: Line 156-157: I am asking for clarification of the text regarding the reduction of gene expression.

RESPONSE: RA-XII (20-40 mg/kg) dose-dependently reduced the expression of lipid synthesis related genes (FASN, SCD) and proteins (Fig. 6D-F). Fatty acid synthase (FASN) synthesizes palmitate from acetyl CoA and malonyl CoA. Palmitate and stearate are subsequently desaturated by the stearoyl CoA desaturase (SCD). This issue has been addressed on page 7 (lines 174-177).

Point 6: Line 190-192: I am asking for precise specification of the topic in the field of gene overexpression and lipid biosynthesis.

RESPONSE: In fact, overexpression of lipogenic enzymes is reported as a common characteristic of many cancers, and inhibition of different enzymes within the fatty acid biosynthetic pathway can block cancer cell growth. Among them, SCD and FASN are two important enzymes during cell de novo lipogenesis. The significant roles of SCD and FASN in cancer development have been well established in the past. Elevated expression of SCD and FASN in cancer cells is related to markedly worse prognosis in many human cancers, including liver cancer.  The SCD inhibitor BZ36 and FASN inhibitor C75 have shown anti-tumor effects in pre-clinical xenograft model (page 8, lines 216-218; page 9, lines 219-223).

This manuscript is a resubmission of an earlier submission. The following is a list of the peer review reports and author responses from that submission.

Round 1

Reviewer 1 Report

This article could be accepted for its publication because of its soundness and scientific quality.

The results indicated that RA-XII strongly inhibited tumor growth and lipogenesis (triglyceride and lipid droplets) in HepG2 cells, and the expression of key factors involved in lipogenesis (SREBP, SCD, FASN) was also obviously downregulated. Further investigation showed that the anti-tumor effects of RA-XII were attenuated by SREBP knockdown. Moreover, RA-XII downregulated the expression of SCAP, an upstream regulator of SREBP, and siRNA of SCAP prevented its restrained effects on tumor growth and lipogenesis. In addition, the in vivo experiment showed that RA-XII strongly restrained the lipogenesis and growth of liver tumor in nude mice xenograft model. Taken together, these results indicate that RA-XII suppresses the liver cancer growth by inhibition of lipogenesis via SCAP-dependent SREBP suppression. The findings reveal the potentials of RA-XII to be used in the novel therapeutic approach in treating liver cancer.

This is an interesting piece of information related with RA-II, that can be the basis for more in-depth studies.

Author Response

Point 1: This is an interesting piece of information related with RA-II, that can be the basis for more in-depth studies.

RESPONSE: Thanks and we will continue studying the anti-tumour effects and the underlying mechanisms of RA-XII in the further investigation.

Reviewer 2 Report

In this manuscript the authors investigated the underlying mechanism of the RA-XII, one of cyclopeptide glycosides isolated from Rubia 20 yunnanensis, as an anti-tumor molecule. The results indicated that RA-XII strongly inhibited tumor growth and lipogenesis (triglyceride and lipid droplets) in HepG2 cells, and the expression of key factors involved in lipogenesis (SREBP, SCD, FASN). The authors showed that the anti-tumor effects of RA-XII were attenuated by SREBP knockdown. Moreover, RA-XII down-regulated the expression of SCAP, an upstream regulator of SREBP, and siRNA of SCAP prevented its restrained effects on tumor growth and lipogenesis. In addition, in vivo experiment showed that RA-XII restrained the lipogenesis and growth of liver tumor in a nude mice xenograft model. The authors suggest that reagents inhibiting lipogenesis such as RA-XII, may be used in a novel therapeutic approach for treating liver cancer.

Lipogenesis plays indeed a critical role in the growth and metastasis of tumors and thus became an attractive target for anti-tumor drugs. Thus, searching for lipogenesis inhibitors is important and a real need, especially for plant derived compounds that are a well-recognized source for known anti-cancer compounds.

Major comments:

The whole study was performed on HepG2 liver cancer cells. However, to establish a compound for a future possible treatment for liver cancer; the authors must perform some of the basic experiments on additional liver cancer cells (there at least 3 other known human liver cancer cells in the recognized cell banks).

As RX-II is dissolved in 20mM DMSO; are control HepG2 cells (0uM RX-II) treated with the same final concentration of DMSO as those added with the compound at the highest concentration? Please detailed this information as its not clear from the manuscript.

Moreover, to prove that RA-XII has specific effects on cancer cells and is not toxic to normal cells, the compound must be tested for non-specific activity and its effects, at least on cell growth, must be tested on normal fibroblast and lymphocytes. The therapeutic window should be tested and determined.

Figs. 1B&1C (mainly figure 1C); impossible to read at this size; size of these figures should be enlarged to enable evaluation of the figures!

As in all the rest of the experiments the concentration of RX-II is expressed in uM; Fig. 1B should be drawn also at this scale. In addition, its not clear how was the IC50 value determined (2.29uM) as it seems from the curve that its not correct!; please check the graph and determine the correct IC50 value (see also Fig. 3C in the presence of 5uM RX-II; No siRNA).

Moreover, the growth inhibition curve is reaching a value of not more than about 70% inhibition on an established liver cancer cell line; hard to see how such compound will have an anti-cancer effect in vivo (see also the comments below regarding its anti-cancer effect in vivo).

Fig. 1C&D: The authors show that RX-II induced a dose dependent G2/M cell cycle arrest. The authors already demonstrated that RA-XII has an anti cancer effect on HepG2 liver cancer cells and claimed in their pervious publication (ref. no. 22 in the manuscript) that RA-XII suppresses protective autophagy for enhancing apoptosis through AMPK/mTOR/P70S6K pathways in HepG2 Cells. In such a case, one should expect an increase in the sub-G1 phase of the cell cycle. Was this measured at all? What is the possible explanation for those two observations? The authors also should comment and combined the results of both studies; do they fit regarding to the mechanisms involved? This should be compared at least in the discussion section.    

Fig. 3D-not readable; please enlarge size of the Fig.

Section 2.3: “SREBP-1 suppression is involved in RA-XII-induced cell death…”.

Please specify which SREBP-1 was examined-SREBP1a or SREBP-1c. Was SREBP-2 also tested? What were the results?

In vivo effect of RX-II: Please indicate if treatment was thrice weekly for the two weeks duration of the experiment. Although some anti-tumor effect is evident; the experiment was terminated after a very short period (15days post start of treatment; Fig. 5). This is a too short time to follow tumor growth and a real effect of the RX-II compound. Mice at each group should be kept for a follow-up, post treatment; to follow tumor growth. In addition, to follow side effects and possible toxicity of the treatment; mice organs should be removed and pathologically examined.

Minor corrections

There are some English editing corrections that needed to be done.

In Methods: please indicate 13,200Xrpm as ?xg.

Author Response

Major comments:

Point 1: The whole study was performed on HepG2 liver cancer cells. However, to establish a compound for a future possible treatment for liver cancer; the authors must perform some of the basic experiments on additional liver cancer cells (there at least 3 other known human liver cancer cells in the recognized cell banks).

RESPONSE: Thanks and we have not revised for the following reasons: (1) In our previous study, the results indicated that RA-XII exhibited cytotoxic activities in various tumor cell lines, including HepG2, SMMC-7721 and BEL-7402 liver cancer cell lines (Fan et al., Bioorg. Med. Chem. 2010, 18, 8226-8234). (2) HepG2 cells have always been used as model for mechanism study on other anti-tumor compounds (Choi et al., Arch. Pharm. Res. 2015, 38, 691-704). Thus, we focused on HepG2 cells in the present study.

Point 2: As RX-II is dissolved in 20mM DMSO; are control HepG2 cells (0uM RX-II) treated with the same final concentration of DMSO as those added with the compound at the highest concentration? Please detailed this information as its not clear from the manuscript.

RESPONSE: This issue has been addressed in the Method section on page 9 (line 232 -235).

Point 3: Moreover, to prove that RA-XII has specific effects on cancer cells and is not toxic to normal cells, the compound must be tested for non-specific activity and its effects, at least on cell growth, must be tested on normal fibroblast and lymphocytes. The therapeutic window should be tested and determined.

RESPONSE: In growing tumours, energy supply is mainly provided by lipids coming from de novo synthesis. To research the regulating effects of RA-XII on de novo lipogenesis, all the liver cancer cells were cultured in lipid deficiency medium (page 9, line 246-247). However, the normal cells such as fibroblast and lymphocytes must absorb lipids from the environment, so we didn’t compare the anti-proliferation effects of RA-XII on different cells.

Point 4: Figs. 1B&1C (mainly figure 1C); impossible to read at this size; size of these figures should be enlarged to enable evaluation of the figures!

RESPONSE: Size of these figures has been enlarged as suggested by the reviewer.

Point 5: As in all the rest of the experiments the concentration of RX-II is expressed in uM; Fig. 1B should be drawn also at this scale. In addition, its not clear how was the IC50 value determined (2.29uM) as it seems from the curve that its not correct!; please check the graph and determine the correct IC50 value (see also Fig. 3C in the presence of 5uM RX-II; No siRNA).

RESPONSE: Thanks and Figure 1B has been edited according to the reviewer’s suggestion. Results are described on page 2 (line 73). We are so sorry for the mistake.

In Figure 3C, cells were transfected with SREBP-1 siRNA or NC siRNA for 24 h, followed by treatment with RA-XII. Since the transfection process affected cell growth, the inhibition rate induced by RA-XII treatment was not the same as Fig. 1B.

Point 6: Moreover, the growth inhibition curve is reaching a value of not more than about 70% inhibition on an established liver cancer cell line; hard to see how such compound will have an anti-cancer effect in vivo (see also the comments below regarding its anti-cancer effect in vivo).

RESPONSE: In our previous study, the results indicated that RA-XII exhibited cytotoxic activities in HepG2 cells cultured in normal complete medium. The growth inhibition curve is reaching a value of more than 70% (Song et al., Molecules, 2017, 22, 1934).

Point 7: Fig. 1C&D: The authors show that RX-II induced a dose dependent G2/M cell cycle arrest. The authors already demonstrated that RA-XII has an anti cancer effect on HepG2 liver cancer cells and claimed in their pervious publication (ref. no. 22 in the manuscript) that RA-XII suppresses protective autophagy for enhancing apoptosis through AMPK/mTOR/P70S6K pathways in HepG2 Cells. In such a case, one should expect an increase in the sub-G1 phase of the cell cycle. Was this measured at all? What is the possible explanation for those two observations? The authors also should comment and combined the results of both studies; do they fit regarding to the mechanisms involved? This should be compared at least in the discussion section.  

RESPONSE: This issue has been addressed in the discussion section on page 9 (line 218 -222).

Point 8: Fig. 3D-not readable; please enlarge size of the Fig.

RESPONSE: Size of the figure has been enlarged as suggested.

Point 9: Section 2.3: “SREBP-1 suppression is involved in RA-XII-induced cell death…”.

Please specify which SREBP-1 was examined-SREBP1a or SREBP-1c. Was SREBP-2 also tested? What were the results?

RESPONSE: We just examined the regulating effects of RA-XII on total SREBP-1 in the present study. The specified effects of RA-XII on different SREBP proteins remain to be investigated in the future.

Point 10: In vivo effect of RX-II: Please indicate if treatment was thrice weekly for the two weeks duration of the experiment. Although some anti-tumor effect is evident; the experiment was terminated after a very short period (15days post start of treatment; Fig. 5). This is a too short time to follow tumor growth and a real effect of the RX-II compound. Mice at each group should be kept for a follow-up, post treatment; to follow tumor growth. In addition, to follow side effects and possible toxicity of the treatment; mice organs should be removed and pathologically examined.

RESPONSE: We are so sorry for our carelessness. Treatment was thrice weekly for the two weeks duration of the experiment and this issue has been addressed in the Method section (page 11, line 318). As shown in Figure S1, RA-XII treatment for two weeks did not cause significant side effects such as, mortality, lethargy and pathologic change of organs. The results have been described on page 7 (line 163). In addition, the long period effect of RA-XII has been evaluated in another project.

Minor corrections

Point 1: There are some English editing corrections that needed to be done.

RESPONSE: Results has been edited as suggested by the reviewer.

Point 2: In Methods: please indicate 13,200Xrpm as ?xg.

RESPONSE: 13,200 rpm approximately equals 11600 g with our centrifuge and this issue has been indicated in the Method section (page 10, line 262).

Reviewer 3 Report

Manuscript entitled “RA-XII suppresses the development and growth of 2 liver cancer by inhibition of lipogenesis via SCAP-3 dependent SREBP suppression”. This manuscript from Wang Y  and colleagues were trying to identify the role of plant-derived compound RA-XII on lipogenesis and tumor growth of liver cancer cells. Authors were conducted both in vivo and in vitro studies to identify the inhibitory effect of RA-XII on cancer cell growth and define the mechanism of SCAP-dependent SREBP suppression during lipogenesis. Although this result is interesting, there was a recent study has demonstrated that the role of RA-XII on autophagy (PMID: 29137114) in liver cancer and they have shown RA-XII impacts on same HepG2 cells. This finding indicates that specificity of RA-XII drugs and its role in disease. Authors should discuss the exact role of RA-XII in cancer.

Author Response

Point 1: Comments and Suggestions for Authors

Manuscript entitled “RA-XII suppresses the development and growth of 2 liver cancer by inhibition of lipogenesis via SCAP-3 dependent SREBP suppression”. This manuscript from Wang Y and colleagues were trying to identify the role of plant-derived compound RA-XII on lipogenesis and tumor growth of liver cancer cells. Authors were conducted both in vivo and in vitro studies to identify the inhibitory effect of RA-XII on cancer cell growth and define the mechanism of SCAP-dependent SREBP suppression during lipogenesis. Although this result is interesting, there was a recent study has demonstrated that the role of RA-XII on autophagy (PMID: 29137114) in liver cancer and they have shown RA-XII impacts on same HepG2 cells. This finding indicates that specificity of RA-XII drugs and its role in disease. Authors should discuss the exact role of RA-XII in cancer.

RESPONSE: This issue has been addressed in the discussion section on page 9 (line 218 -222).

Round 2

Reviewer 2 Report

Response to Point 1:

Checking the the reference indicated by the authors to prove that RA-XII has cytotoxic activities in various tumor cell lines, including HepG2, SMMC­7721 and BEL­ 7402 liver cancer cell lines it turns out that it has an effect on SMMC7721 cells but has NO EFFECT on BEL­7402 liver cancer cells; not enough to answer on the the comment! In addition, although HepG2 is used in many studies; additional liver cancer cell lines must be tested and evaluated.

Response to Point 3:

The response is not acceptable! Normal fibroblasts and lymphocytes can easily survive

6 hr. in 1% lipoprotein deficient serum (as described in the method section for experimental conditions); thus the effect of RA-XII must be tested on normal human fibroblasts and lymphocytes to determine its non-specific toxicity!

Response to Point 6:

Even an effect of more than 70% growth inhibition (how much more); is still a concern for RA-XII potency as an effective anti cancer compound.   

Again; "In addition, the long period effect of RA­XII has been evaluated in another project" as a response, is not acceptable.

Reviewer 3 Report

None